# The Entropy of Digital Texts—The Mathematical Background of Correctness

**DOI:** 10.3390/e25020302

**Published:** 2023-02-06

**Authors:** Mária Csernoch, Keve Nagy, Tímea Nagy

**Affiliations:** Faculty of Informatics, University of Debrecen, Kassai út 26., 4028 Debrecen, Hungary

**Keywords:** word processing, errors, correction, modification, formatting, communication entropy, text-entropy

## Abstract

Based on Shannon’s communication theory, in the present paper, we provide the theoretical background to finding an objective measurement—the text-entropy—that can describe the quality of digital natural language documents handled with word processors. The text-entropy can be calculated from the formatting, correction, and modification entropy, and based on these values, we are able to tell how correct or how erroneous digital text-based documents are. To present how the theory can be applied to real-world texts, for the present study, three erroneous MS Word documents were selected. With these examples, we can demonstrate how to build their correcting, formatting, and modification algorithms, to calculate the time spent on modification and the entropy of the completed tasks, in both the original erroneous and the corrected documents. In general, it was found that using and modifying properly edited and formatted digital texts requires less or an equal number of knowledge-items. In information theory, it means that less data must be put on the communication channel than in the case of erroneous documents. The analysis also revealed that in the corrected documents not only the quantity of the data is less, but the quality of the data (knowledge pieces) is higher. As the consequence of these two findings, it is proven that the modification time of erroneous documents is severalfold of the correct ones, even in the case of minimal first level actions. It is also proven that to avoid the repetition of the time- and resource-consuming actions, we must correct the documents before their modification.

## 1. Introduction

Millions of digital texts have been created since word processors appeared on the market, and handling these documents became one of the most popular digital production-activity. However, Johnson [1] called attention to his findings as early as late the 90s that:


*“…it is believed that a very little knowledge, a skimpy overview, is sufficient. The value of limited study of word processing applications is rather doubtful. In fact, a little bit of knowledge about word processing may be almost useless—and a quick overview is certainly not sufficient to realize most of the overwhelming benefits of using computers for writing.”*


He went even further, claiming that:


*“In order to take advantage of the powers of word processing requires considerable skill in its use. Those who understand only a little about word processors will probably employ them in a linear fashion like an expensive typewriter and compose in exactly the same way they would on a typewriter because they simply do not know how to use the sophisticated editing features of a word processor.”*


What surprising is that the quality of digital texts has not improved considerably during the past years, and the mentioned misconceptions regenerate and intensify the problems. Johnson [1] only mentions the complexity of word processors and users’ lack of knowledge; however, we also must pay attention to the content and the structure of the texts, which is more complex and demanding than the technical details of using word processors. It is said that building football stadiums does not make football, having a telescope does not make astronomy. In a similar way, owning hardware and software does not make informatics, or computer sciences, which is clearly expressed both in science [2] and industry [3,4].


*“When major new machinery comes along—as computers have—it’s rather disorientating.”*
[2]


*“Relying on technical innovation alone often provides only temporary competitive advantage.”*
[3]


*“Society has reached the point where one can push a button and be immediately deluged with technical and managerial information. This all very convenient, of course, but in one is not careful there is a danger of losing the ability to think. We must remember that in the end it’s the individual human being who must solve the problems.”*
[4]

Consequently, the quality of word-processed documents is questionable due to the circulating misconceptions and the ill-use of these rather complex applications [5,6,7,8,9,10,11,12,13,14,15,16,17,18,19,20], which can be proven by thorough analyses of the available documents and rather inefficient modification processes. More and more word-processed documents are publicly available on the internet and circulate in closed networks. On one hand, the erroneous documents encourage untrained end-users to copy and follow these examples, and they use them as references. On the other hand, it allows researchers to build their corpora for various error detection studies on the subject. Considering the available word processors and their popularity at the time of writing the present paper, we selected the most widely used program and its documents for further analysis, which was Microsoft Word and documents with DOC and DOCX extensions [21]. One further reason for choosing Microsoft Word was that it allows users to seamlessly present non-printing characters on the interface which—despite its credible popularity—is crucial in the editing process of digital texts [22,23,24,25].

The question is whether self-taught [26], overconfident [27,28,29], tool-centered end-users (non-professional authors and/or editors of documents) [30,31,32,33,34,35,36,37,38,39,40,41,42,43], or those who raise doubts about the quality of word-processed texts have merit [5,6,7,8,9,10,11,12,13,14,15,16,17,18,19]. To find proof, instead of relying on legends and folktales [44,45], we must find facts by using the results of educational research connected to the subject [44,45,46,47,48,49,50,51,52,53]. Being aware of the low efficiency of word processing activities, we launched a research project to find an objective measuring system and define the entropy of erroneous and correct digital texts, what we call text-entropy.

Our previous research in the field of didactics of informatics has already revealed that in general, education performs a crucial role in developing students’ computational thinking skills [54], especially in handling digital texts [5,6,7,8,9,10,11,12,13,14,15,16,17,18,19]. It is also found that the cognitive load of handling digital text-based documents in a “user-friendly” GUI is so high [55] that novel methods and approaches are required to be able to cope with this complex problem.

Considering the role of education, the complexity of digital texts, the algorithm-driven, deceptive “user-friendly” word processing programs, and the millions of erroneous documents circulating either in closed communities (corporate, school, training group, etc.) or on the internet available for everyone, it is found that we are in great need of approaches and methods that can handle errors. Surprisingly, educational research has paid little attention to learning from errors [56], with a few exceptions [57,58,59]. Primarily sports, where ignoring errors would lead to serious injuries and underperformance in competitions [60,61,62,63,64]. Not less surprising is that the profit-oriented “real-world” and their production systems also pay minimal attention to errors. However, the Toyota Production System claims that they must bring problems to the surface, make them visible, and go to work immediately on countermeasures [4]. One of the principles of the Toyota Way is that:


*“We view errors as opportunities for learning. Rather than blaming individuals, the organization takes corrective actions and distributes knowledge about each experience broadly. Learning is a continuous company-wide process as superiors motivate and train subordinates; as predecessors do the same for successors; and as team members at all levels share knowledge with one another.”*


However, most of these principles, and educational and/or training research consider students’, employees’, athletes’, etc., errors committed by themselves which must be corrected as soon as possible to avoid further damages.

Based on this concept, but using already existing erroneous documents and teaching-learning materials as examples, the Error Recognition Model (ERM) was introduced [7,11,12]. We use the very same model to define the entropy of digital texts (text entropy) and decide whether a text is correct or not.

### 1.1. Error Recognition Model

The Error Recognition Model (ERM) [7,11,12] would be a solution for increasing the effectiveness of digital text management, since it has been proven more efficient and effective than the widely accepted tool-centered approaches, and the circulating misconceptions. Both are primarily rooted in the published teaching-learning materials, or rather tutorials, the disadvantages of tool-centered computer courses [30,31,32,33,34,35,36,37,38,39,40,41,42,43], and the believes of overconfident but ignorant end-users [27,28,29].


*“… the significant advantages of word processing are available exclusively to those who are proficient in the use of the hardware and software; they will be inaccessible to those who have only a little understanding of word processing. A stand-alone computer skills course (taught by a school or by a computer dealer) may not be the best means to teach substantial knowledge of word processing; examples and practice will inevitably be simulated and artificial, and there will be little motivation to fully understand the applications.”*
[1]

It is proven that applying ERM [12] in the teaching-learning process has advantages which the widely accepted tool-centered, low-mathability approaches miss [2,65,66,67,68,69,70]. These advantages are due to “real-world” texts used as examples and the consideration of cognitive load at each step of method. The erroneous “real-world” texts perform a crucial role in motivation, where students claim proudly that they can do better. By keeping the cognitive load at bay [44,71,72,73,74], we can make countermeasures immediately and develop students’ computational thinking skills step-by-step. Consequently, we can avoid unrepairable damage when students and end-users create, edit, and modify their own or somebody else’s documents.

#### 1.1.1. Properly Formatted Text

The ERM accepts the definition of the properly formatted text [7,11], and, based on it, builds up a teaching-learning approach that focuses on the text, instead of the word processing tools and interfaces.

The definition of the properly formatted text has two constrains which are the following:The text fulfils the requirements of printed documents (quantitative requirements, and errors detailed in Section 1.1.2).The content is indifferent to modification (qualitative requirements, and errors detailed in Section 1.1.2)—the document is editable, but applied changes must be limited to the correct form of those actions that the user originally intended.

#### 1.1.2. Error Categories

The focuses of ERM are the recognition of errors in digital texts (e.g., finding, being aware of, avoiding, and correcting them), learning how to avoid them, and transferring this knowledge and skills to other types of digital texts (e.g., from word processors to presentations and web pages, and the other way around). ERM teaches that we must be aware of errors, we must learn how to handle and avoid them to improve the quality of our works [3,4,56,57,58,59]. One of the primary principles of ERM is that errors are user-friendly tools in handling digital contents, as they are in the real world [3,4,56,57,58,59].

In accordance with the definition of the properly formatted text and due to the extremely high number of errors in digital texts, they must be categorized. Considering the terminology of both natural and artificial languages in connection with computers, errors are listed in two hypernym, then three and three hyponym categories [11].

Quantitative error categories (recognizable in any printed or electronically displayed form, can be seen by the reader or viewer of the document):a.syntactic (the grammar of the text);b.semantic (the content of the text);c.typographic (the appearance of the text) [75,76,77,78].Qualitative error categories (recognizable in editable digital form only; therefore, invisible to the target audience, and seen exclusively by the author(s) and other participants handling the original or the modified file):a.layout (the arrangement of characters and objects of the text);b.formatting (applying formatting commands);c.style (handling—applying, defining, reformatting—styles).

It is not rare that one error category triggers another or a third. In these cases, primary and secondary categories are applied to the errors, depending on which triggers the other. For example, in Figure 1 Line 19 has a manual hyphenation which is a layout error. However, the manual hyphenation (with the Enter and Space characters in the middle of the word ‘parental’) creates syntactical and/or semantical errors (with the separate words ‘pa’ and ‘rental’).

#### 1.1.3. The Application of ERM

The application of ERM in an educational environment consists of three major steps depending on the tools used by students/teachers can be unplugged (UP), semi-unplugged (SUP), and digital (D) [12].

The first step is error recognition in two clearly distinguishable phases:a.The marking, explaining (writing), and later categorizing the quantitative errors of the text in its printed form (paper, picture, or PDF). In this phase, the only source is the printed text (UP). This is the form of the document which readers and viewers see.b.The marking, explaining (writing), and later categorizing the qualitative errors of the text in its editable (digital) form. In this phase, the primary source is the editable, opened document (if available with the turned-on **Show/Hide** button to clearly present the non-printing characters [22,23,24,25]). The secondary source is the printed document where notetaking takes place. To distinguish the two types of error hypernyms, different colors are used to mark the errors. The coloring system would also serve students in practicing, rehearsing, and catching up (SUP).The second step is correcting the errors of the document. In this phase, a correction algorithm is set up whose primary concept is to start with the errors of the widest range (domain), and finish with the least significant. The algorithm also considers correction with the **Replace…** command or with macros when repeated errors must be corrected (e.g., multiple empty paragraph marks, tabulator and/or space characters, repeated formatting) (D).The third phase is the formatting of the text which, similar to correction, is based on an algorithm. The steps of the algorithm are primarily decided focusing on the range of the planned formatting commands, starting from those of the widest range to the least significant (D).

We must note at this point that there are cases where some steps of the third phase (formatting) must be conducted in the second phase (e.g., modifying page setup, setting **Normal** style).

The steps of ERM reveal that the focus is on the text [12]. On one hand, the model supports the readers by making the text as legible and understandable as possible. On the other hand, it helps in avoiding and correcting qualitative errors, and making the text ready for effective modification and formatting restricted only to the intention of the author/editor. In this scenario, the text-editing application, the use of the GUI performs only a secondary role, meanwhile building proficiency in the ERM text-handling process.

Unlike traditional course books, teaching-learning materials, and tutorials [30,31,32,33,34,35,36,37,38,39,40,42,43], this concept does not start teaching text management through:operating system instructions (this knowledge should have been brought in through knowledge-transfer, and if not, then it should be introduced and practiced somewhere else) [30,32];toying with font-face appearances [33,34,35,36];new features of the application [30];the introduction of the GUI objects [30];typing text.

Instead of these, in ERM real world texts with real contents are provided during the teaching-learning process, which idea is in complete accordance with the main concept of:Informatics Reference Framework [79,80];TPCK (Technological Pedagogical Content Knowledge) [81,82,83];Polya’s concept-based problem solving approach [65];Kahneman’s thinking fast and slow theory [71] along with the cognitive load theory [72];Hattie’s results on teaching effectively [73];Chen’s Meaning System Model [66].

Considering the combination of these theories, the teaching-learning materials can be adjusted to the students’ (in general, and end-users) age, background knowledge, interests, school subjects, work requirements, etc.

### 1.2. Detecting End-User Activities—ANLITA

To record what actions end-users perform in digital texts, our research group developed a dedicated logging application, named ANLITA (Atomic Natural Language Input Tracking Application). In the present research, ANLITA is primarily used to log all the keyboard actions and to screen-record the complete editing process conducted in Word documents. The program outputs two files:a text file containing all the keyboard events;a video file, in which the entire text handling process was recorded.

Using these two files, the complete text handling process can be retraced, the techniques and algorithms (if there was any) applied to the text can be revealed. Based on this data, an objective measurement system can be set up which might help us to provide information on the quality of the text, more precisely how efficiently the message of the natural language digital text is communicated.

## 2. Sample

For the present study, three documents of different content and length were selected, each carrying both quantitative and qualitative errors:A single page proposition of a Grade 7 student voicing requests of their class for better conditions in the school (for short: *medicine*) (Figure 1). This document includes text-content only. (Line numbering shown in Figure 1 was added for referencing purposes in the present paper.)A teacher’s three-page long test paper (Figure 2). This document includes a combination of text-content, pictures, other graphical objects, and two table-imitations (for short: *frenchfood*).An equation cut from a longer document created by a senior pre-service teacher of mathematics and library information (for short: *equation*) (Figure 3).

These documents are from our private collection, gathered from students and colleagues rendered unidentifiable.

In general, our study checked for the flexibility of these documents, e.g., how efficiently the intended modifications can be applied to them. Consequently, our primarily concerns were the qualitative errors [7,11]. To identify those, the visibility of non-printing characters perform a crucial role in the analysis, so all recordings and figures were created with the **Show/Hide** button turned-on [22,23,24,25] and some with the text boundaries of the document sections made visible.

### 2.1. Document: medicine

The *medicine* document contains several layout errors in the form of multiple Space and Enter characters and Enter characters at the end of the lines (Lines 4, 5, 6, 12, 18, and 19). In general, these characters were meant to create left indentation and vertical spacing between paragraphs, and right alignment in Line 30. We could not explain the multiple Space characters in Line 34. There is a formatting and style error in Line 12 which is unreasonable (style Heading 1 with the **Keep with next** paragraph formatting, reformatted to match the appearance of the main body of the text). Line numbers were not part of the original document, we turned those on to enable us referring to the location of the errors in a more convenient and reliable way.

### 2.2. Document: frenchfood

The original *frenchfood* document is a three-page long text created by a teacher with testing purposes. The length of the document depends on the settings of the **Normal** style, but it was meant to be three pages long. The first page (Section 1) has **Portrait,** while the second and third pages **Landscape** orientation (Section 2) set up.

The document is burdened:with layout errors in the form of multiple Tabulator, Space, and Enter characters,a mixture of layout and formatting errors on the graphical objects (lines and textboxes both in the body and the fake headers),style and formatting errors on fonts and paragraphs.

### 2.3. Document: equation

The content of the *equation* document is a portion of a senior pre-service teacher’s lesson plan. Figure 3 clearly presents the document creator’s attempted manipulation of the text (Ben-Ari [5] calls this bricolage). Instead of inserting proper math formulae into the document using the readily available built-in equation editor, the author fabricated various text-positioning tricks to construct these formula-lookalikes. The errors in this piece of text are the following.

The first formula starts with a manual numbering. (Later in the analysis, the manual numbering was ignored since it does not perform any role in the editing of the equations.)Numerators and denominators are typed in separate paragraphs.Vincula are imitated by applying underline character formatting on the numerators.Minus signs are substituted with underscore characters.Plus signs and whole number are received a subscript character formatting.Equal signs are imitated by blanks (Space characters) with double-underline formatting applied.Space characters are used to horizontally position numerators and denominators.

This document has one further characteristic we must mention; the overconfident author blamed MS Word not allowing equation sign at end the of the lines (marked by a footnote in Formula 1, and by the imitation of footnotes in Formulas 2 and 3) (Figure 3) [27,28,29].

## 3. Methods

On the selected documents, a rigorous analysis was conducted to reveal their structures and to detect their errors. This is primarily an unplugged and/or semi-unplugged process according to ERM [12].

Following the steps of ERM, in the second phase, the errors of the original documents were corrected, then the proper formatting took place. Finally, both to the original and the corrected-formatted documents, modifications were applied to measure and compare the entropy of the documents.

We must note that all these steps can be repeated by anyone interested. However, the steps detailed in the present paper were conducted by two of the authors. They both are professionals in text management, which helped them to set up a normalized algorithm to each text and to minimize the time assigned to the algorithm steps. Consequently, less experienced end-users repeating all these steps might provide different time and probability. However, these differences not necessarily deteriorate the entropy as the limit to how effectively one can handle a digital text, to how effectively one can communicate the outcome of a text handling process (Section 6).

### 3.1. Correction of the Documents

Considering the content, each piece of text is unique. However, the analyses revealed that the primary concern our analysis is the improper use of non-printing characters. Since we were not present at the creation of the documents, we do not know whether the authors had these turned on or off [22,23,24,25]. Regardless of the authors’ awareness of the non-printing characters and our lack of this information, all the processes applied to the texts during the study are conducted and recorded with the turned-on **Show/Hide** button. We can use this method to disambiguate whether characters or formatting are used to set up an appearance, which perform crucial role both in the analyses and the documentation of the analyses, and later in the dissemination of the findings.

A correction-algorithm was set up for to the *medicine* and *frenchfood* documents, along which their correction was executed. Beyond correcting the errors of a document, the algorithm serves our research to set up a lower bound on the average size (number of instructions and time) of each modification that end-users can apply to a text. With this method, we can define correction-entropy and the number of bits a text requires to be corrected.

These steps of the algorithms are presented in in Section 5, along with the time spent on each step.

### 3.2. Formatting of the Documents

Following the correction of the documents, all the three texts were properly formatted which includes the modification of the **Normal** style. Furthermore, where it was necessary, minor adjustments were applied. This happened in the case of the *frenchfood* document where some of the pictures on the second page must be resized to fit the content of the three rows on one page.

In a similar way to correction, a lower bound on the average size of each formatting is set up. The steps of the algorithms are presented in in Section 5, along with the time spent on each step.

The aims of the correction and formatting processes were to set up the entropy of the documents and find an objective measuring system which can distinguish erroneous texts from their properly formatted counterpart.

### 3.3. Correction, Formatting, and Statistics

In the following, the methods of the correction and formatting of the *medicine* and the *frenchfood* documents are detailed. Beyond the algorithms of these processes, the statistics and other graphical tools provided by the software (GUI) are added for giving more explanatory details.

#### 3.3.1. Document: *medicine*

The corrected *medicine* document consists of 8 paragraphs (Figure 4, right) compared to the 34 paragraphs in the original (Figure 4, left).

The text boundaries (Figure 4) reveal that MS Word is able to recognize each individual paragraph of the document, including the empty ones. However, the document statistics calculated and shown by the application are manipulated (Figure 5) by leaving out the empty paragraphs, despite that they perform a crucial role in the text-handling processes. According to the statistics (Figure 5), the original medicine document has 14 paragraphs, because MS Word counts those paragraphs only where at least one alphabetical/numerical/special printable character content is present.

In the case of the *medicine* document, the aim of the correction was to clear all the unnecessary and incorrect characters and formats of the document. While in the formatting phase, we wanted to apply proper formatting steps to set up a correct document as similar in appearance to the original one as possible.

#### 3.3.2. Document: *frenchfood*

The comparison of the boundaries of the erroneous (Figure 6) and the corrected document (Figure 7), and the statistics of the original and the correct *frenchfood* documents (Figure 8) reveals that the empty paragraphs are not counted in the statistics. Furthermore, it can be concluded that only those paragraphs are counted which have printable characters, and, in that respect, neither figures nor drawn shapes count as paragraph-content.

According to the MS Word statistics (Figure 8), in the original document there are 15 paragraphs: 10 + 1 on Page 1 (P1), and 3 + 1 on Page 2 (P2). On P1, the paragraph of the orange does not count (there is no printable character in this paragraph), while both of the text boxes do (one on P1 and the other on P2). In the corrected document, there are 10 paragraphs on P1 and 3 on P2. However, the text boundaries reveal that the original *frenchfood* document is loaded with uncounted empty paragraphs, and there are two unnecessary text boxes.

Figure 9 presents the corrected and formatted *frenchfood* document. In this document, the first paragraph of P1 is moved to the header, a table was created on each page, and the pictures and food names were put side by side on P1, and names were arranged below the pictures on P2. Furthermore, on P2 a line (bottom border with paragraph domain) was added to each food name to mimic the appearance seen on the original document.

In the correction phase of the analysis, the aim was to create two tables holding all the pictures and food names. On P1, we kept the original order of the pictures and names, while on P2, the pictures and names got rearranged into matching pairs.

### 3.4. Modification of the Documents

Beyond calculating correction- and formatting-entropy of the documents, we also wanted to calculate the modification-entropy. In this context, modification means that either content is added to the document, or a formatting is performed.

The modification of the documents was planned and executed in accordance with the content and the characteristics of the texts. Considering these criteria, the following modifications were applied to both the original and the corrected documents (Table 1).

#### 3.4.1. Document: *medicine*

In the *medicine* documents two modifications were applied (both to the original and the corrected). In the first analysis, two words were added to the second paragraph (as counted in the correct document) (Figure 10), while in the second analysis, the font size of the paragraph was changed (Figure 11).

#### 3.4.2. Document: *frenchfood*

In the original *frenchfood* document on P2, ten figures were arranged into three rows. The first two rows consist of four and four pictures, while the third row contained only two pictures. To fill in this gap, two pictures and two food names were added to the content of P2 (Figure 12).

The second modification to the *frenchfood* document was the changing of the font size. Figure 13 presents the results of this modification both in the original (left) and in the corrected (right) documents. While in the original document further amendments were required to match the pictures and the food names, in the corrected document no additional steps were required.

#### 3.4.3. Document: *equation*

Introducing the equation editor in word processors is considered difficult, thus it is seldom taught in schools and trainings. In general, only advanced or special classes deal with the subject. However, to measure the difficulty and the complexity of creating equations, we launched several subjective and an objective test. In the subjective tests, the equation editor was introduced in a Grade 7 math class. In the subjective tests, the equation editor was introduced in a Grade 7 math class. Before we introduced the equation editor, these students had already used touch screen displays and their fingers to enter equations. During the testing period where the equation editor was introduced and used, both the activities and the results of the students, along with the time spent on the computer were observed. Furthermore, the eligibility of the formulae were considered.

To find objective measures, we first applied our planned modifications to the formula-lookalikes in the erroneous document following the jiggery-pokery method of the original author, then to the properly constructed equations. (Figure 3).

The modifications performed in this objective measuring process were:calculating the common denominator of the fractions (Figure 14),adding a new fraction to the equation (Figure 15),solving the substitution problem by simplifying the equation (Figure 16).

The final step of the test was to measure the entropy of both solutions, to see how much data is required to put on the communication channel to handle equations (formulas).

## 4. Setting Up the Theory

### 4.1. Connection to Communication Entropy

In Shannon’s original paper [84,85], the probabilities of events were calculated primarily from their relative frequency (Equation (3)). In our case, we were obliged to find other sources to calculate the probabilities. We found that the duration of events can substitute the frequency and serve as the basis for calculating text-entropy.

The correction, formatting, and modification phases of the documents are digital processes, and, as such, can be logged with ANLITA. As it was mentioned earlier (Section 1.2) ANLITA produces two files for each monitored session (what we call a modification task):a text file logging the keyboard activities and their timestamp,a video file with the entire modification process recorded.

The recorded files can be used to calculate the time spent on the events (atomic steps). The sum of these event-times provides the time spent on the modification task (Equation (1)). Using these durations and the total time, we are able to calculate the probability of a modification task (Equations (1) and (2)) [84,85,86,87,88,89,90].

### 4.2. Calculating the Entropy of Digital Texts—Text-Entropy

The text-entropy can originate from three different type of tasks:formatting,correction (in erroneous documents),modification.

To all types, the same method is applied to calculate their text-entropy. Based on the algorithm of the task, events (atomic steps) are set up, and to each event, a time is assigned (*t_k_*, recorded in the logging process and measured in seconds with two decimals). The sum of these time values provides the time of the task (*t*) (Equation (1)).
(1)$$t=t_{1}+\cdots +t_{n}=\sum_{k=1}^{n}t_{k}$$

Based on the duration of the events (*t_k_*) and duration of the task (t), the probability of each event is calculated (*p_k_*) (Equation (2)). The sum of these probabilities is 1 (Equation (3)). Calculating the probability of the events from the time assigned to them, allows us to distinguish text-entropy from information-entropy, while still holding to a common theoretical background.
(2)$$p_{k}=\frac{t_{k}}{t}$$
(3)$$p_{1}+\cdots +p_{n}=\sum_{k=1}^{n}p_{k}=1$$

From these probabilities (*p_k_*) the information content of the events (*I_k_*) were calculated (Equation (4)). (The choice of a logarithmic base is explained in Section 4.4)
(4)$$I_{k}\left(p\right)=-{log}_{2}p_{k}$$

The final step of calculating text-entropy is presented in Equation (5). The sum of the products of probabilities and information contents allows us to quantify the entropy of a task (*E*). In other words, we are able to tell how many bits of data are required to be put on the communication channel to conduct the planned task (Equation (5)).
(5)$$E\left(Χ\right)=\sum_{k=1}^{n}p_{k}\cdot I_{k}=-\sum_{k=1}^{n}p_{k}\cdot \log_{2}p_{k}=\sum_{k=1}^{n}E_{k}$$

To provide text-entropy as an integer, one further step is applied, where *E* is rounded up (*Eb*) (Equation (6)).
(6)$$Eb=\left[E\left(Χ\right)\right]+1$$

### 4.3. The Characteristics of Data in Text-Entropy

One further question is what kind of data can be considered when text management is discussed. Shannon, in his original schematic diagram of a general communication system, called the data being put on the communication channel “information source”. In our case, the situation is the same. To handle digital texts, end-users must have sources, which enable them to conduct the planned actions. Calculating the entropy of an action does not specify these sources. However, our knowledge connected to text management can identify two sources with certainty. These are education, and the user interface of the word processor which is recently a graphical interface (GUI). Further sources can be considered, but they are beyond the scope of the present paper.

Considering the possible sources of information (the data being put on the communication channel), we must keep in mind that the analyses presented in the paper were conducted by the professionals of our research group. In the case of general end-users, measuring the entropy of the selected modification tasks might require further steps, items, and concerns, which might increase the number of bits to be put on the channel (further discussed in Section 6).

### 4.4. The Choice of a Logarithmic Base

According to Shannon [84,85], any base can be selected to describe the entropy. Since in computer sciences base 2 is the most widely accepted, we decided to use the same for calculating the amount of data required to complete a modification task.


*“If the number of messages in the set is finite then this number or any monotonic function of this number can be regarded as a measure of the information produced when one message is chosen from the set, all choices being equally likely. As was pointed out by Hartley the most natural choice is the logarithmic function. Although this definition must be generalized considerably when we consider the influence of the statistics of the message and when we have a continuous range of messages, we will in all cases use an essentially logarithmic measure.”*



*“The choice of a logarithmic base corresponds to the choice of a unit for measuring information. If the base 2 is used the resulting units may be called binary digits, or more briefly bits, a word suggested by J.W. Tukey. A device with two stable positions, such as a relay or a flip-flop circuit, can store one bit of information. N such devices can store N bits, since the total number of possible states is 2^N^ and log_2_2^N^ = N. If the base 10 is used the units may be called decimal digits.”*


## 5. Results

To calculate the entropy of a document we decided to take into consideration:first level modifications, which are fundamental actions (typing, changing font size, and inserting pictures),the correction of the document to avoid carrying the inefficient modification processes,the formatting of a document.

If the document is properly edited and formatted, the correction and the formatting of the document is left out from both the text-management process and the analysis. Furthermore, if the original document carries errors, modifications were applied in both the original and the corrected documents to be able to compare the messages which the two forms of the document carry.

One further technique in the process of measuring the entropy of digital texts is the grouping of repeating steps. With this method, for example handling multiple Space, Tab, and Enter characters are not logged character by character. This method allows space for various solutions handling these repetitions (e.g., deleting with replacement, selecting by blocks or double clicks, etc.), reduces the steps of the algorithm to a reasonable size, and the time assigned to these steps can be measured in seconds. The results of the analyses are shown in the following tables (Tables in Section 5) where the steps are presented as modification events (Column **Algorithm**), to which the duration, the information content and the entropy values are assigned(**Time (tk)**, **Ik,** and **Ek**, respectively).

Considering the formatting of the documents, in the testing process these steps were performed only in the correct version of the documents to avoid the multiplication of errors.

Following these concerns, the next sections provide the results of how the algorithms were set up and the entropy assigned to these algorithms.

### 5.1. Document: medicine–Correction

The algorithm of the correction of the *medicine* document is presented in Table 2. In general, it is found that two bits of data (*Eb*) are required to put on the communication channel to correct this document. The crucial steps of the algorithm are the following:the recognition of the extra Space and Enter characters,how to delete these characters,how to remove all the font and paragraph formats to clear the typographic and formatting errors,recognizing and correcting syntax errors.

Clearing all the errors of the *medicine* document took around 230 seconds. To remove multiple Space and Enter characters two methods were recorded:selecting blocks and deleting them,using replacement (**Replace…** command), where two Space or Enter characters were substituted by one and handling the line opening and closing Space characters.

In this document, there was no significant difference between the times spent on the two methods.

After correcting the *medicine* document, it has eight paragraphs, and there is no unnecessary character left (Figure 4 right, and Figure 5 right). In the next step, the formatting of the corrected document took place.

### 5.2. Document: medicine–Formatting

The algorithm of the formatting, along with the time spent on the atomic steps (events) and the information content of the steps of the *medicine* document is presented in Table 3. To format this document three bits of data (*Eb*) must be put on the channel.

At first sight (and originally meant to be by the author), this document is an easy to handle little thing. However, the power of this document is underestimated. The entropy of the formatting reveals that the knowledge pieces required by this document are extremely demanding. The three bits of data needed to format this document indicate that firm background knowledge is necessary to complete the task. Calculating the entropy of this “easy” task explains the errors of the original document. Furthermore, it proves that minimal guidance [74] is not enough to teach fundamental word processing. A lot more data must be put on the channel than course books and other teaching-learning materials suggest [33,34,35,36,37,38,39,40].

### 5.3. Document: medicine–Modification

The modification of the *medicine* document has four stages:inserting two words into the original (Table 4) and the corrected (Table 5) documents (Figure 10),increasing the font size of one paragraph in the original (Table 6) and the corrected (Table 7) documents (Figure 11).

In the erroneous text, typing two words into a paragraph requires additional steps to restore the appearance similar to the original, imitating the left indent. These additional steps are typing and deleting Space and Enter characters. With this method, three bits of data must be put on the channel to reach the expected arrangement (Table 4). However, we must mention a further side effect of this solution: whenever the content of the text is changed, adjusting the text to the fake ident must be repeated, which is an extremely time- and resource-consuming process.

Completing the “insert two words” task in the corrected document requires only one bit of data. In this case, purely the intention of the end-user (the typing) needs to be completed, no collateral activities are imposed.

The comparison of the entropy of the two corrections reveals that handling an erroneous document is much more demanding than a correct one. In this case, typing two words required almost three times more time, and around three times more data put on the channel in the erroneous document.

Further typing and content modification would have the same requirements with a linear increase in time. However, we calculated the time and the entropy of formatting of the *medicine* document (Table 3) and found that these tasks required three bits of data. The comparison of the entropy of typing in the erroneous document is around as much as the entropy of the formatting. We can conclude that if end-users want to work effectively in a digital text, the content first need to be correctly edited and formatted. In any other cases, the word processing activities are disadvantageous, we lose time, money, and resources.

Increasing the font size of a paragraph in the original (erroneous) document requires four bits of data (*Eb*). This can be explained by the collateral actions imposed on the user to realign the fabricated left indent. These actions are beyond the modification intention of the end-user (which was simply to just change the font size) and based on the entropy this is quite demanding.

In the corrected document, changing the font size of the selected paragraph took less than 5 s, and one bit of data must be put on the channel.

### 5.4. Document: frenchfood–Correction

The *frenchfood* document is supposed to have two tables, one on Page 1 (P1) and the other on Page 2 (P2). Instead of using two actual tables, the original document is loaded with fiddly attempts to imitate them. Due to the high number of layout errors of the *frenchfood* document, the correction took more than 380 s, and four bits of data (*Eb*) were needed (Table 8). This shows that recognizing the errors of a document is quite demanding and should be handled more consciously during the teaching-learning process; primarily, to avoid these errors, secondarily to be able to correct them before any modification is applied to the document.

The steps of the correction algorithm reveal that after deleting the multiple Space, Tab, and Enter characters, the most important knowledge pieces are the inserting and formatting tables.

### 5.5. Document: frenchfood–Formatting

The formatting of the corrected document took around 100 s, and three bits of data must be put on the channel (Table 9). End-users intended to modify this document must apply fundamental font and paragraph formatting in tables, handle borders in table with paragraph domain, format header with positioned tabulators, and set up the location of the header.

### 5.6. Document: frenchfood—Modification: Inserting Picture and Text

In the modification phase of the research, two pictures of food and their names were inserted into the empty cells of T2 on P2 (Figure 12). The modification was conducted both in the original (erroneous) and the corrected and formatted documents. The steps of these processes are presented in Table 10 and Table 11.

It is found that the modification of the original document is more demanding than that of the correct one. It required more than twice as much time and knowledge pieces in the erroneous document than in the corrected one. In both cases, the pictures needed to be inserted and the words needed to be typed, which took around the same amount of time. Adjusting the size of the pictures was conducted in both documents without any serious alteration in time. However, in the original document additional steps were required to adjust the pictures and the words (one below the other and horizontally centered). These steps took up time.

### 5.7. Document: frenchfood–Modification: Changing Font Size

Changing the **Normal** style with the font (Arial) and font size (32 pt) of the original and the corrected documents show remarkable differences. While in the original document it took around 300 s (Table 12), in the corrected document, only around 20 s were needed (Table 13). Furthermore, it is also remarkable that changing the font size and then adjusting the pictures and text require four bits of data (Table 12), while in the corrected document one bit is needed (Table 13). In general, we can conclude that handling erroneous documents is more difficult, complicated, and demanding than handling correct ones.

### 5.8. Document: equation–Modification

The *equation* document is different both from *medicine* and *frenchfood* in the sense that it cannot be corrected. The formulae of the *equation* document are so poorly designed that our research group gave up on its correction but created the correct formulae using the built-in equation editor of the word processor. Figure 17 presents the paragraphs of the erroneous and corrected *equation* documents, with the text boundaries made visible.

In this phase of our research, the original *equation* document was modified by adding new fractions to the original formula in accordance with the following concept:finding the common denominator of the fractions (Figure 14, Table 14 and Table 15),solving the problem and substituting the parameters with constants (Figure 16, c = −12, d = 18) (Table 16 and Table 17),adding new fraction to the formula (Figure 15, Table 18 and Table 19).

Calculating the common denominator of the fractions (Figure 14) requires five bits of data in the erroneous (Table 14), and four in the correct (Table 15) document. Furthermore, the time spent on the actions is almost four times more in the erroneous document than in the correct one (Table 14 and Table 15). In the correct document, creating the common denominator requires nothing else but positioning the cursor, deleting the old value and typing the new. In the case of the erroneous formulae, additional time is required to adjust the position of denominators and numerators with Space characters, and formatting the equal sign and the vincula with further adjustments.

The following step of the analysis was to imitate the solving of the presented problem in the original lesson plan (Figure 16 and Figure 17). In this phase, several atomic actions were performed, for example inserting new fractions, typing numerators and denominators, simplifications, and calculations. Similar to calculating the common denominator, beyond the required elements in the formulae, collateral adjustments and formatting were imposed. Considering the number of knowledge pieces required to complete this task, in both cases five bits of data must be put on the channel (Table 16 and Table 17). However, the time spent on the task is 2.6 times more in the erroneous document (Table 16 and Table 17).

In the third test of the *equation* documents, a new fraction was added to the existing one (Figure 15). In this case, the number of bits which must be put on the channel in the erroneous (Table 18) and the correct (Table 19) documents was 3 and 2, respectively. The time spent on this modification is 3.3 times more in the erroneous (Table 18) than in the correct (Table 19) document.

## 6. Discussion

### 6.1. Changing Duration and Adding a New Atomic Step

In the previous section, the entropy of correction, formatting, and modification tasks in the three sample documents were detailed, and the results presented using the newly introduced concept of “text-entropy”. In the discussion section, we focus on the question, how the entropy of a task changes when:more time is spent on an already existing atomic step (event), ora new atomic step is inserted.

It is found that changes in the duration of atomic steps is a subcategory of the insertion of a new atomic step. Consequently, this latter, more general modification is analyzed. Figure 18 presents the modification of the correct *frenchfood* document. An additional fictitious atomic step is added to the steps listed in Table 11 (with a t = 100 s duration mapped in Figure 18). The time assigned to the action is in the range of [1, 1000] with a 1 s difference.

It is found that in both cases the entropy increases until we reach a certain amount of time (*t_max_*), which is understood as further knowledge pieces are added as input data. However, when *t_max_* is reached the entropy starts to decrease, this also means that without adding further atomic steps to the process, the time conceals further events. We call this concept hidden or time-concealed data.

### 6.2. Where We Stand

Millions of ill-treated Word documents circulate both in closed communities and on the internet providing bad samples. In both cases, the available documents include educational materials meant to teach word processing and offering exercises on the subject. In general, it is found that these matters and problems are invisible to IT professionals, corporate managers, and researchers in both the fields of information systems and computer education. 

In Figure 19, Figure 20 and Figure 21 erroneous Word tutorials and exam papers are presented. The samples show that even educational documents do not fulfill the requirements of the properly structured and formatted text and are set up as bad examples.

These exam papers repeat exactly the kind of errors our research analyzed. The examples prove the negligent text-handling habits and demonstrate errors that exist in the teaching–learning process of text-editing in general.

The selected errors are the following:empty paragraphs (Figure 19, Figure 20, Figure 21 and Figure 22),manual numbering (Figure 20, Figure 21 and Figure 22),horizontal positioning within a paragraph using Tab characters (Figure 19 and Figure 20),centering paragraph with Tab character (Figure 19),empty paragraphs before a manual page break (Figure 21 and Figure 22),incorrect use of expression: open an application instead of start/run (Figure 19).

### 6.3. Building Corpora

To our knowledge, open-source datasets of Word documents do not exist. Another problem, considering corpora, is that the copyright issues are not defined. Consequently no one knows under what conditions such corpora can be created and used. This explains our choice of samples in the paper, where the Word documents are from our private collection.

Author-data is rarely available, consequently there is no place to ask for permission.There are no designated data banks of corpora through which copyright issues might be handled.This entire process is a black box.Documents found on the public internet are not guaranteed to stay there.

## 7. Conclusions

This paper presents how text-entropy can be calculated from text handling processes, which are primarily formatting, correction, and modification (including the creation of a text). The word processing tasks can be algorithmized, broken into events (atomic steps) to which durations of time can be assigned. These lengths of time serve as sources of probabilities, bases of information entropy, and, from now on, of text-entropy.

In the present study, three erroneous MS Word documents were selected to demonstrate how text-entropy can be applied to real-world digital texts. In this process, we have to build the algorithm of the tasks, measure the time spent on the events, and finally calculate the entropy of the completed tasks.

In the comparison of the erroneous and the correct documents, the text-entropy can be calculated from the entropy of modification tasks. In the present analysis, first level modifications [19]—typing, changing font size, inserting picture—are conducted in the *medicine* and the *frenchfood* documents, and inserting formulae in the *equation* documents. Furthermore, the entropy of correction and formatting of the original (erroneous) documents must be taken into consideration when calculating text-entropy. These pieces provide data about the complexity of documents, and on how correct or how erroneous they are.

We found that modifying a properly structured and formatted digital text is less demanding than its erroneous counterpart. It means that less data must be put on the communication channel than in the case of erroneous documents. The analysis also revealed that not only the quantity of the data is lower, but the quality of the data (the required knowledge pieces) is higher. As the consequence of these two findings, it is proven that the modification time of erroneous documents is severalfold that of the correct ones, even in the case of minimal first level tasks. It is also proven that to avoid the multiplication of time- and resource-consuming tasks, we need to correct the documents before their modification.

With the concept and the methods provided in the present paper, we were able to find an objective measurement—the entropy of digital texts—that can describe the quality of documents. Based on this value, it can be told how correct or how erroneous a digital text-based document is.

The analyses can be further extended to several directions. First, we must emphasize that the algorithms were set up and the steps were conducted by the members of our research group, who are professionals in text management and handling word processors. Less experienced end-users might need more time and different algorithms to conduct the modifications, which can generate results different from those presented here. Furthermore, it is clear from the results that end-users, primarily students must be taught how to design and create properly structured and formatted digital texts, and how to work effectively with digital texts by using complex word processors. As the results of our analyses revealed, the focus of education should be realigned from the interface of an application to the content and structure of the digital texts.

It is also found that the presented approach for calculating text-entropy provides a method that can be followed both in educational and industrial (business/office/firm) environments to test the correctness of documents and reveal their discrepancies, if there is any. The aim of introducing this concept and testing method is to reduce the use and waste of both human and machine resources in the process of handling digital text-based documents.

## Figures and Tables

**Figure 1 entropy-25-00302-f001:**
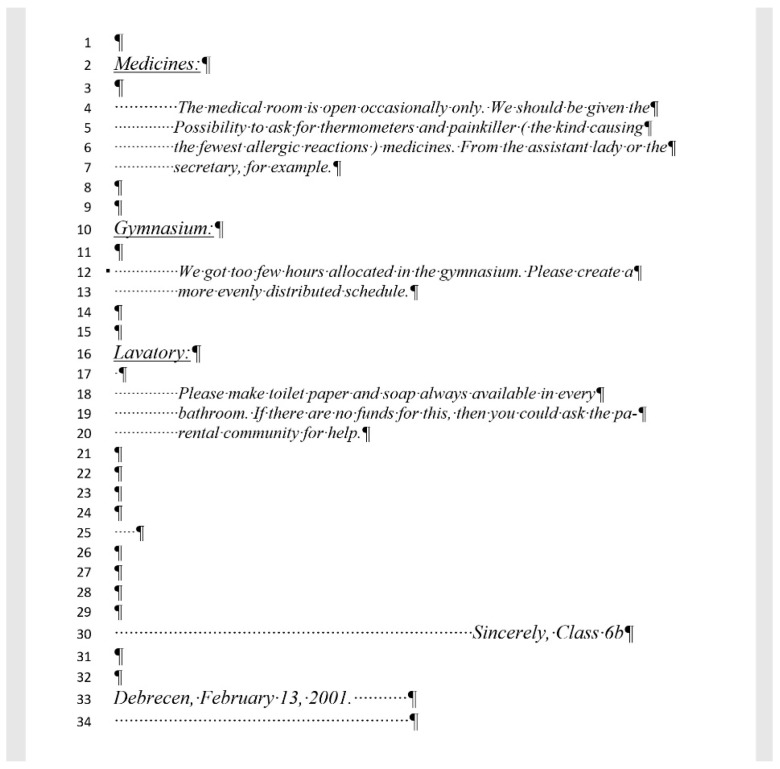
The translated version of the original Hungarian single page document, named *medicine*.

**Figure 2 entropy-25-00302-f002:**
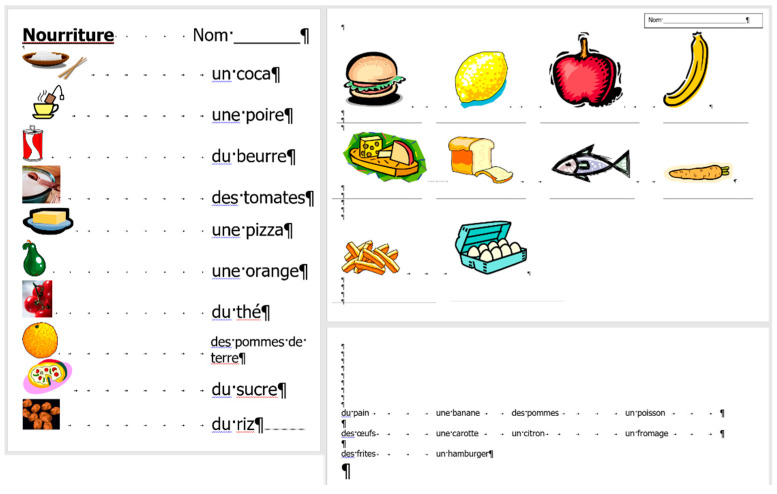
The original three-page long *frenchfood* document.

**Figure 3 entropy-25-00302-f003:**
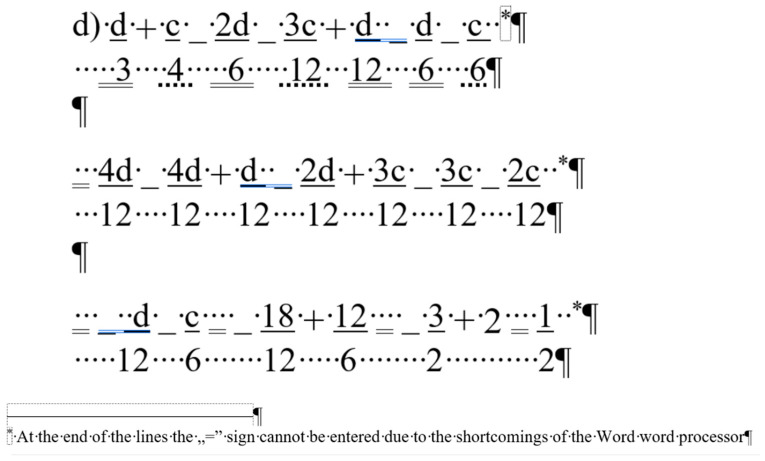
The original formula-lookalikes from a senior pre-service teacher’s lesson plan, the document named *equations*.

**Figure 4 entropy-25-00302-f004:**
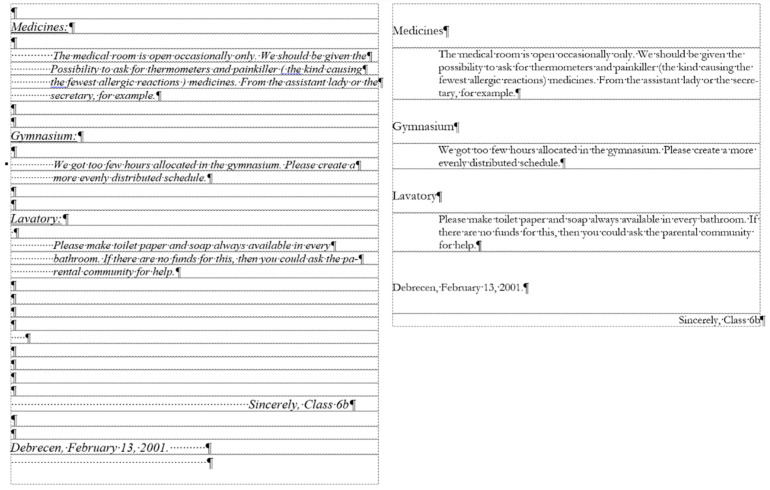
The paragraphs of the original and the corrected *medicine* documents, with the text boundaries made visible.

**Figure 5 entropy-25-00302-f005:**
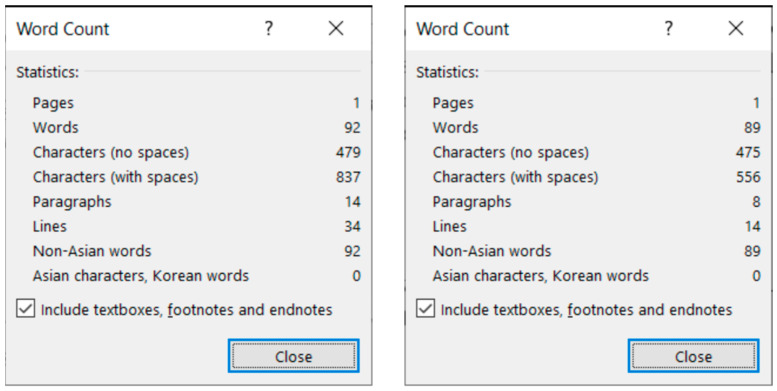
MS Word statistics of the original (**left**) and the corrected (**right**) *medicine* documents.

**Figure 6 entropy-25-00302-f006:**
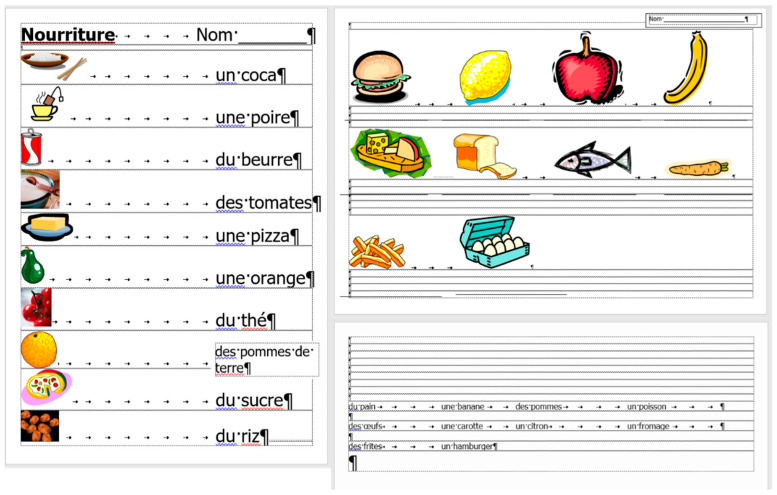
The original three-page long *frenchfood* document with visible text boundaries.

**Figure 7 entropy-25-00302-f007:**
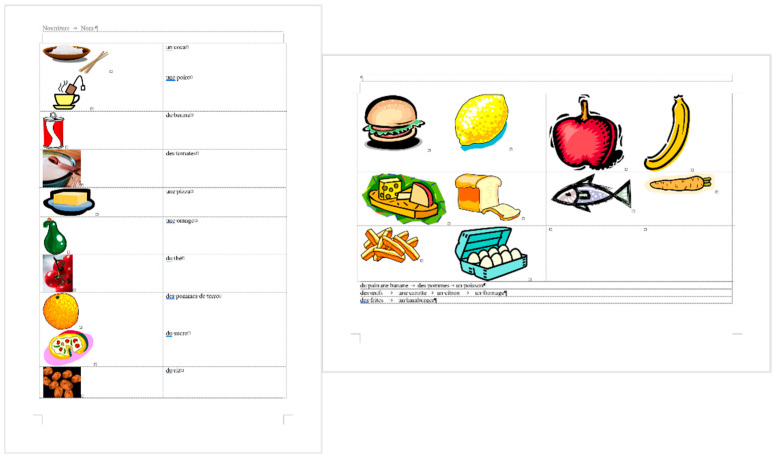
The corrected two-page long *frenchfood* document with visible text boundaries.

**Figure 8 entropy-25-00302-f008:**
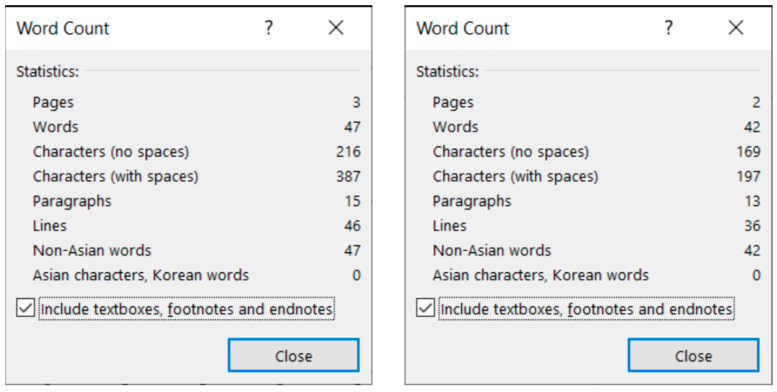
MS Word statistics of the original (**left**) and the corrected (**right**) *frenchfood* documents.

**Figure 9 entropy-25-00302-f009:**
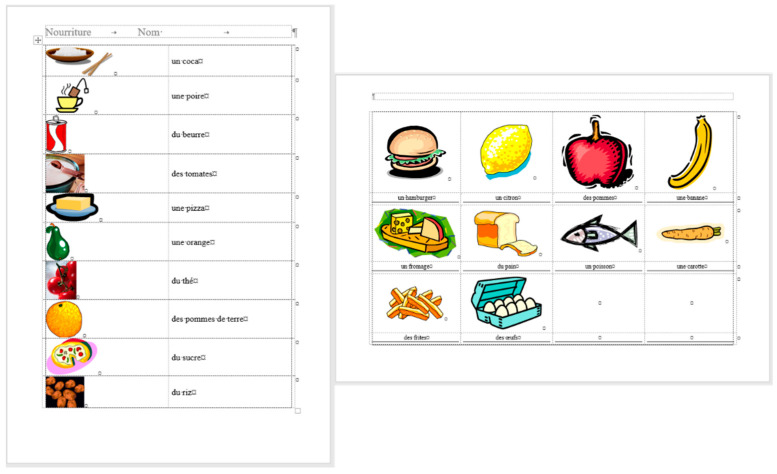
The corrected and formatted two-page long *frenchfood* document with visible text boundaries.

**Figure 10 entropy-25-00302-f010:**
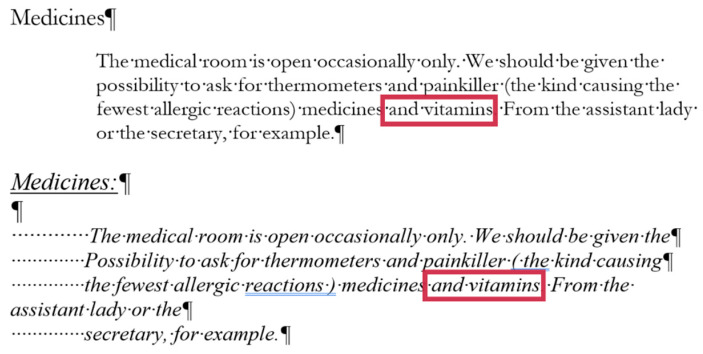
The expression “and vitamins” (red box) is inserted into the corrected (**upper**) and the erroneous (**lower**) *medicine* documents.

**Figure 11 entropy-25-00302-f011:**
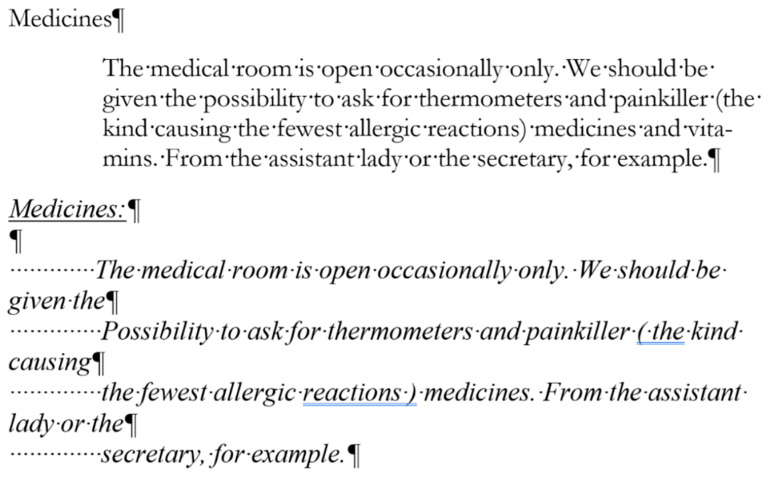
Font size is changed in the corrected (**upper**) and the original, erroneous (**lower**) the *medicine* documents.

**Figure 12 entropy-25-00302-f012:**
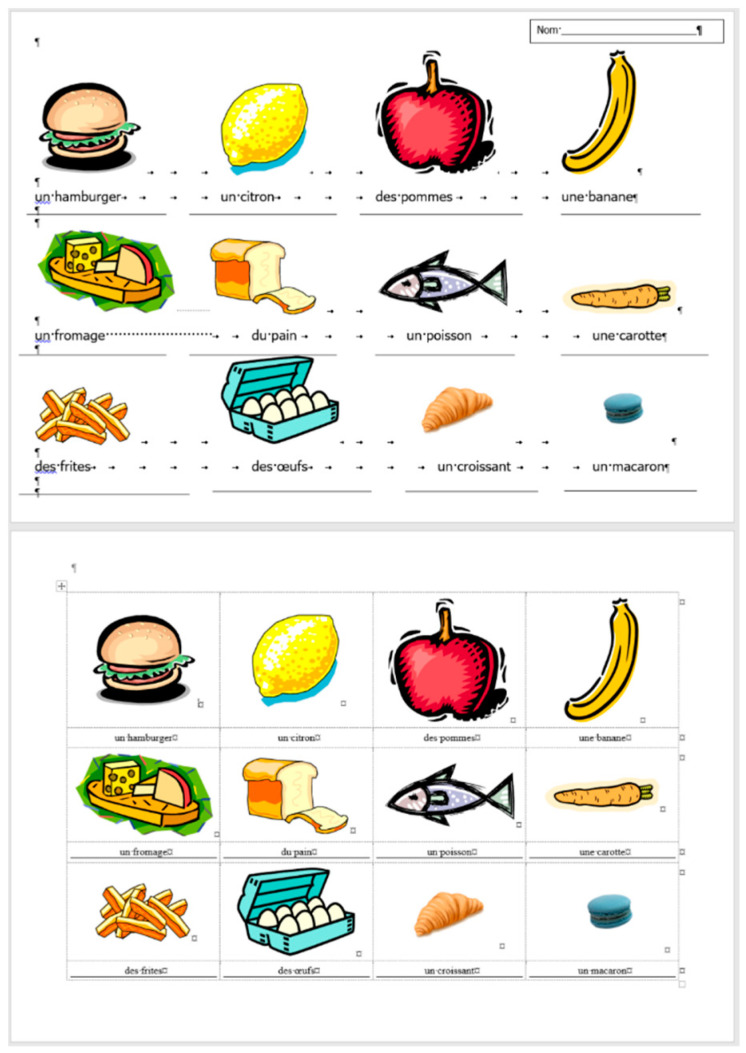
The second page of the original (**upper**) and the corrected (**lower**) *frenchfood* documents.

**Figure 13 entropy-25-00302-f013:**
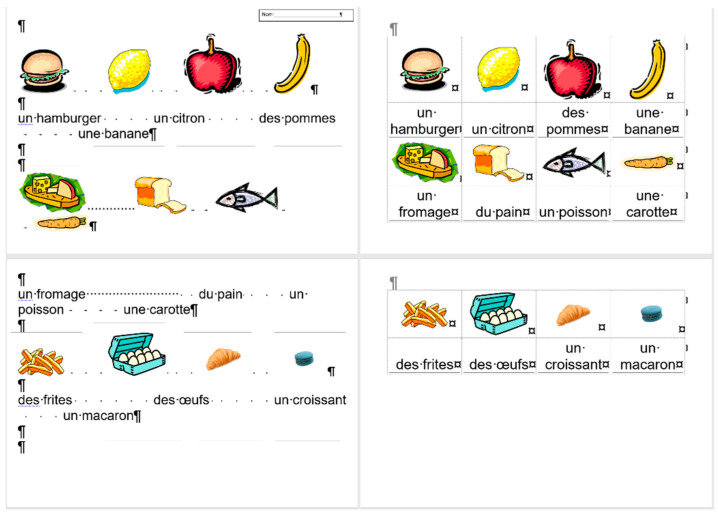
The results of changing the font size in the original (**left**) and the corrected and formatted (**right**) *frenchfood* documents.

**Figure 14 entropy-25-00302-f014:**
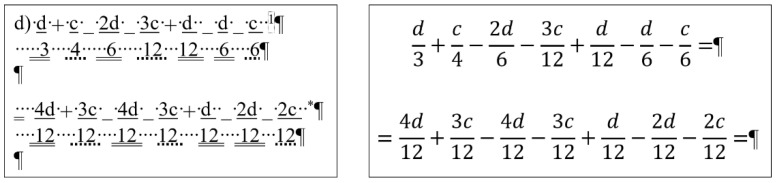
Calculating the common denominator of the fractions in the original (**left**) and in the corrected (**right**) *equation* documents.

**Figure 15 entropy-25-00302-f015:**
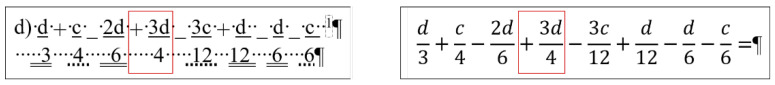
Adding a new fraction (red box) to the original (**left**) and the corrected (**right**) *equation* documents.

**Figure 16 entropy-25-00302-f016:**
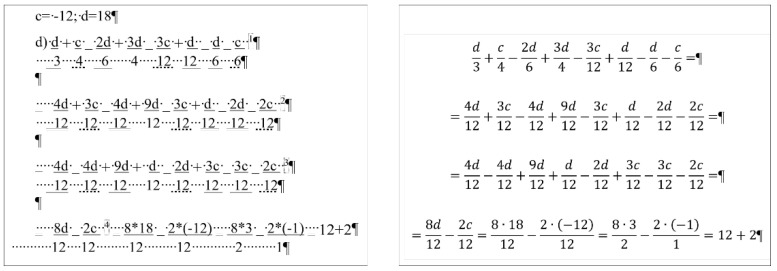
Solving the substitution problem in the original (**left**) and in the corrected (**right**) *equation* documents.

**Figure 17 entropy-25-00302-f017:**
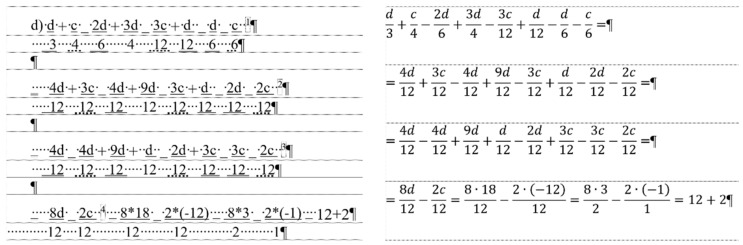
The paragraphs of the original (**left**) and correct (**right**) *equation* documents with the text boundaries made visible.

**Figure 18 entropy-25-00302-f018:**
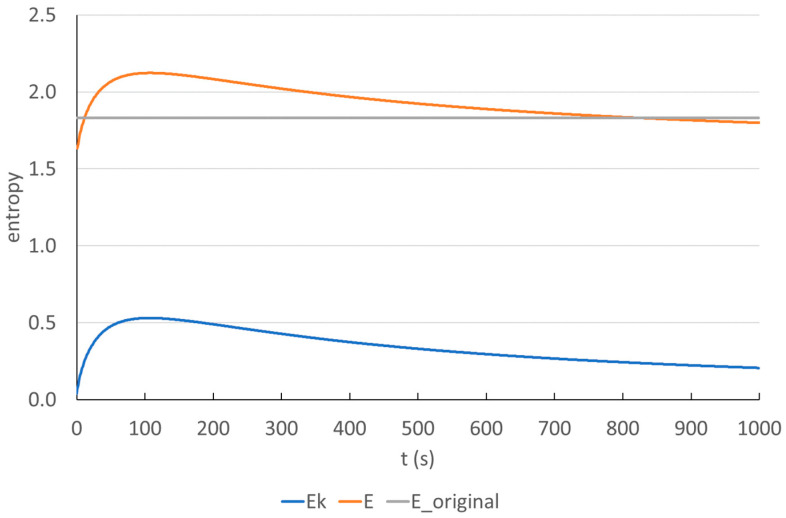
The changes in the entropy of a newly introduced atomic step (*Ek*) and the entropy (*E*) of a modification task in the function of time in the *frenchfood* document (*E_original* = 1.83 (Table 11) indicates the entropy before the modification).

**Figure 19 entropy-25-00302-f019:**
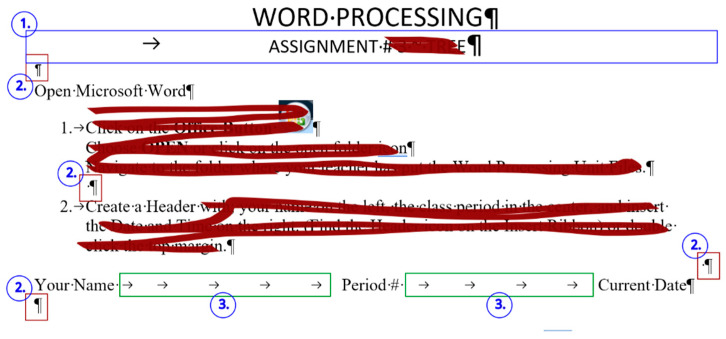
Errors in a Word exam paper.

**Figure 20 entropy-25-00302-f020:**
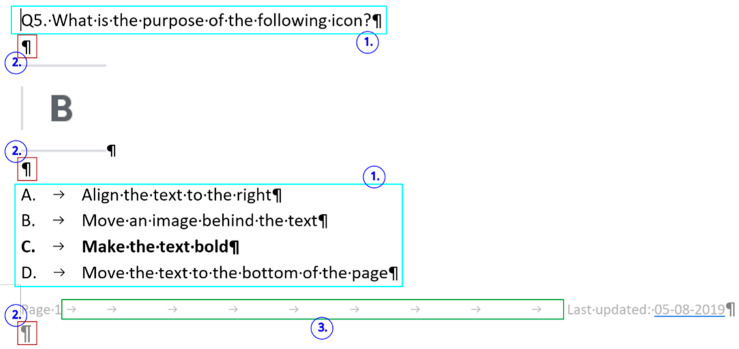
Errors in a Word exam paper.

**Figure 21 entropy-25-00302-f021:**
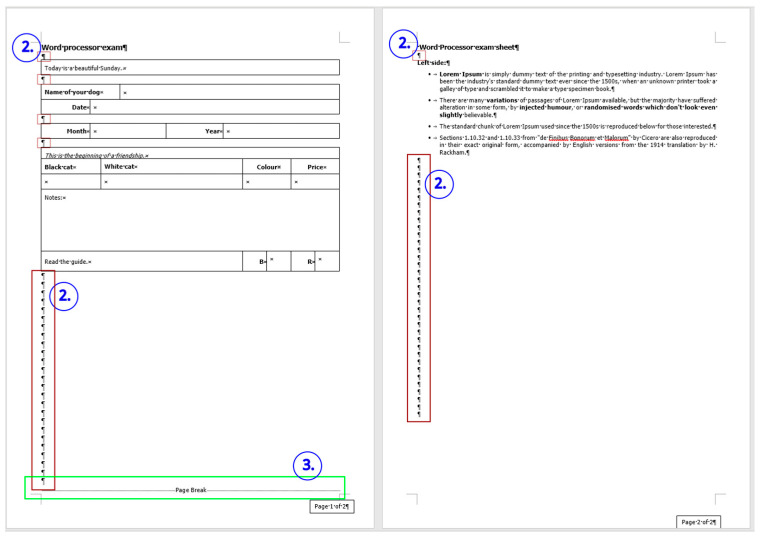
Errors in a Word exam paper.

**Figure 22 entropy-25-00302-f022:**
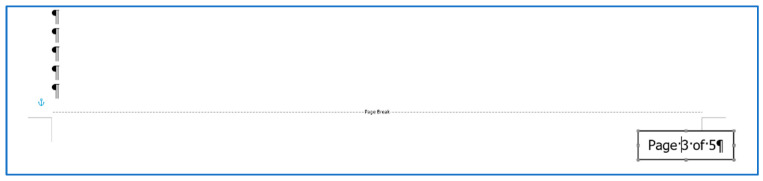
Errors in a Word exam paper.

**Table 1 entropy-25-00302-t001:** Modifications applied to the three sample documents of the study.

Document	Modification1	Modification2
medicine	typing two words	changing font size
frenchfood	adding two pictures and names	changing font size
equation	adding new fractions	simplifying formula

**Table 2 entropy-25-00302-t002:** The steps of the correction of the *medicine* document.

Algorithm	Time (tk)	Ik	Ek
clearing all font and paragraph formatting	5.00	5.5262	0.1199
deleting Space blocks	22.93	3.3263	0.3316
deleting leading Spaces	13.03	4.1431	0.2345
deleting empty paragraphs	25.07	3.1976	0.3485
moving date above signature	2.96	6.2765	0.0810
deleting end of line Enters, hyphenation, and paragraph-closing Space	27.13	3.0844	0.3637
correcting syntax errors	133.95	0.7804	0.4543
**Total**	**230.07**		**1.9336**

**Table 3 entropy-25-00302-t003:** The formatting steps of the *medicine* document.

Algorithm	Time (tk)	Ik	Ek
modifying Normal style	18.10	2.5217	0.4391
left indent and copying	15.89	2.7096	0.4142
titles: Font size, vertical Spacing, and copying	32.12	1.6942	0.5236
date: vertical Spacing	14.00	2.8923	0.3896
signature: right alignment	10.88	3.2560	0.3408
hyphenation	12.95	3.0047	0.3744
**Total**	**103.94**		**2.4817**

**Table 4 entropy-25-00302-t004:** The steps of inserting two words into the original (erroneous) *medicine* document.

Algorithm	Time (tk)	Ik	
positioning the cursor	5.09	2.6941	0.4163
typing	6.86	2.2636	0.4714
typing Enter	5.12	2.6856	0.4174
typing Space	7.01	2.2324	0.4751
deleting Enter	2.86	3.5258	0.3061
deleting Space	6.00	2.4568	0.4475
**Total**	**32.94**		**2.5338**

**Table 5 entropy-25-00302-t005:** The steps of inserting two words into the corrected *medicine* document.

Algorithm	Time (tk)	Ik	Ek
positioning the cursor	5.09	1.2313	0.5245
typing	6.86	0.8007	0.4597
**Total**	**11.95**		**0.9841**

**Table 6 entropy-25-00302-t006:** The steps of changing font size in the original *medicine* document.

Algorithm	Time (tk)	Ik	Ek
selection	1.89	6.0947	0.0892
changing font size	3.03	4.0582	0.2436
typing Enter	1.17	4.7459	0.1769
typing Space	4.80	4.2997	0.2183
deleting Enter	2.98	2.3086	0.4660
deleting Space	4.06	4.3763	0.2107
typing Enter	16.14	3.7267	0.2815
typing Space	3.85	3.9878	0.2514
deleting Enter	6.04	3.1481	0.3551
deleting Space	5.04	3.0051	0.3743
typing Enter	9.02	3.3284	0.3313
typing Space	9.96	4.3140	0.2169
deleting Enter	7.96	5.4028	0.1277
deleting Space	4.02	4.7219	0.1789
**Total**	**79.96**		**3.5219**

**Table 7 entropy-25-00302-t007:** The steps of changing font size in the corrected *medicine* document.

Algorithm	Time (tk)	Ik	Ek
selection	1.89	1.3803	0.5302
changing font size	3.03	0.6993	0.4307
**Total**	**4.92**		**0.9609**

**Table 8 entropy-25-00302-t008:** The correction steps of the erroneous *frenchfood* document.

Algorithm	Time (tk)	Ik	Ek
deleting all font and paragraph formatting	6.14	5.9627	0.0956
changing proofing language to French	16.91	4.5011	0.1988
deleting textbox on P1 and moving the text	36.09	3.4074	0.3211
deleting the textbox on P2	19.02	4.3315	0.2151
deleting lines on P2	32.85	3.5431	0.3040
checking the paper size	51.03	2.9076	0.3875
setting the margins to default	14.98	4.6759	0.1829
deleting multiple Tabs	30.95	3.6290	0.2933
replacing Space characters with a Tab on P2	1.03	8.5383	0.0230
moving the first paragraph of P1 to the Header	7.15	5.7430	0.1072
deleting underscore characters	22.94	4.0611	0.2433
deleting paragraph closing Tabs	42.02	3.1879	0.3498
creating table on P1 (T1)	18.81	4.3475	0.2136
formatting T1	35.09	3.4479	0.3160
creating table on P2 (T2)	12.92	4.8894	0.1650
formatting T2	34.99	3.4520	0.3154
**Total**	**382.92**		**3.7316**

**Table 9 entropy-25-00302-t009:** The formatting steps of the *frenchfood* document.

Algorithm	Time (tk)	Ik	Ek
setting font size in T1	8.93	3.4856	0.3112
setting alignment in T1	3.00	5.0593	0.1517
adding lines (borders) in T2	11.11	3.1705	0.3521
adding vertical space in T2	22.84	2.1308	0.4865
setting alignment in T2	13.16	2.9262	0.3850
borders in T2 (empty lines)	40.99	1.2871	0.5274
formatting Header with positioned tabs, setting header only on P1	38.93	1.3615	0.5299
**Total**	**100.03**		**2.7438**

**Table 10 entropy-25-00302-t010:** The steps of inserting two pictures and names into the original (erroneous) *frenchfood* document.

Algorithm	Time (tk)	Ik	Ek
inserting new paragraphs	9.05	5.4766	0.1230
moving words	99.84	2.0130	0.4987
typing Tabs/Spaces	65.18	2.6282	0.4251
adjusting non-printing characters	25.05	4.0078	0.2491
deleting characters from the end of the document	3.84	6.7134	0.0640
adding pictures	31.95	3.6568	0.2899
adjusting picture size for using only two pages	60.07	2.7460	0.4093
adding food names	33.84	3.5739	0.3001
adjusting with Spaces and Tabs	74.15	2.4422	0.4494
**Total**	**402.97**		**2.8087**

**Table 11 entropy-25-00302-t011:** The steps of inserting two pictures and names into the corrected *frenchfood* document.

Algorithm	Time (tk)	Ik	Ek
moving words to the empty text cells in T2	96.96	0.9319	0.4885
deleting characters from the end of the document	4.04	5.5169	0.1205
adding pictures to the empty picture cells in T2	34.83	2.4090	0.4536
adding food names in the empty text cells in T2	20.13	3.2000	0.3482
adjusting picture size for using only two pages	29.02	2.6722	0.4192
**Total**	**184.98**		**1.8300**

**Table 12 entropy-25-00302-t012:** The steps of changing the font size in the original (erroneous) *frenchfood* document.

Algorithm	Time (tk)	Ik	Ek
changing Normal style	16.00	4.2287	0.2255
clear formatting	6.96	5.4296	0.1260
deleting Tabs on P1	5.05	5.8924	0.0992
modifying textbox1	22.00	3.7693	0.2764
modifying textbox2	16.05	4.2242	0.2260
deleting Tabs on P2	13.02	4.5261	0.1964
arranging lines	51.82	2.5333	0.4376
deleting Tabs and Spaces	30.14	3.3151	0.3331
arranging lines	69.96	2.1003	0.4898
deleting Tabs	31.00	3.2745	0.3384
arranging lines	37.98	2.9816	0.3775
**Total**	**299.98**		**3.1260**

**Table 13 entropy-25-00302-t013:** The steps of changing the font size in the corrected *frenchfood* document.

Algorithm	Time (tk)	Ik	Ek
changing Normal style	12.06	0.6615	0.4182
clear formatting	7.02	1.4431	0.5307
**Total**	**19.08**		**0.9489**

**Table 14 entropy-25-00302-t014:** The steps of calculating the common denominator in the erroneous *equation* document.

Algorithm	Time (tk)	Ik	Ek
copying equation	9.05	4.2466	0.2237
adjusting with Spaces	21.02	3.0308	0.3708
formatting equal sign	8.97	4.2594	0.2224
typing numerator	7.02	4.6130	0.1885
typing denominator	2.95	5.8638	0.1007
adjusting with Spaces	4.11	5.3854	0.1288
typing numerator	12.94	3.7307	0.2810
typing denominator	4.96	5.1142	0.1477
formatting vincula	13.07	3.7163	0.2827
typing numerator	8.07	4.4119	0.2073
typing denominator	2.94	5.8687	0.1004
adjusting with Spaces	10.91	3.9769	0.2526
typing numerator	11.09	3.9533	0.2552
typing denominator	2.93	5.8736	0.1002
formatting vincula	6.97	4.6233	0.1876
typing numerator	3.09	5.7969	0.1043
typing denominator	3.90	5.4610	0.1240
formatting vincula	4.09	5.3924	0.1284
trial and error	14.75	3.5419	0.3041
adjusting with Spaces	11.91	3.8504	0.2669
formatting vincula	7.05	4.6069	0.1891
**Total**	**171.79**		**4.1664**

**Table 15 entropy-25-00302-t015:** The steps of calculating the common denominator in the correct *equation* document.

Algorithm	Time (tk)	Ik	Ek
copying equation	7.05	2.6398	0.4236
typing equal sign	5.01	3.1327	0.3572
typing numerator	5.88	2.9016	0.3883
typing denominator	2.05	4.4218	0.2063
typing numerator	3.05	3.8487	0.2671
typing denominator	2.94	3.9016	0.2611
typing numerator	4.07	3.4324	0.3179
typing denominator	2.93	3.9066	0.2605
typing numerator	4.00	3.4575	0.3147
typing denominator	2.03	4.4360	0.2049
typing numerator	2.04	4.4289	0.2056
**Total**	**43.94**		**3.4655**

**Table 16 entropy-25-00302-t016:** The steps of solving the problem in the erroneous *equation* document.

Algorithm	Time (tk)	Ik	Ek
copying equation	12.58	5.4459	0.1249
adjusting with Spaces	12.58	5.4459	0.1249
formatting equal sign	9.73	5.8163	0.1032
typing numerator	10.68	5.6815	0.1107
typing denominator	4.06	7.0761	0.0524
typing numerator	5.75	6.5765	0.0689
typing denominator	1.95	8.1324	0.0290
adjusting with Spaces	5.81	6.5595	0.0695
typing numerator	10.71	5.6776	0.1109
typing denominator	3.93	7.1253	0.0510
adjusting with Spaces	15.46	5.1487	0.1451
typing numerator	2.97	7.5308	0.0407
typing denominator	1.90	8.1762	0.0283
typing numerator	3.91	7.1324	0.0508
typing denominator	0.95	9.1688	0.0159
formatting vincula	19.48	4.8148	0.1711
copying equation	6.86	6.3200	0.0791
changing order	43.77	3.6471	0.2911
adjusting with Spaces	15.45	5.1496	0.1451
formatting vincula	40.75	3.7504	0.2787
copying equation	32.12	4.0935	0.2398
numerator (calculation)	22.50	4.6073	0.1890
denominator (calculation)	7.65	6.1633	0.0860
adjusting with Spaces	16.64	5.0420	0.1530
numerator1 (substitution)	23.30	4.5565	0.1936
numerator2 (substitution)	1.81	8.2443	0.0272
adjusting with Spaces	9.74	5.8148	0.1033
formatting equal sign	10.85	5.6594	0.1120
formatting vincula	17.48	4.9713	0.1585
adjusting with Spaces	7.72	6.1505	0.0866
simplifying fraction1	28.10	4.2866	0.2196
simplifying fraction2	1.99	8.1040	0.0295
adjusting with Spaces	18.56	4.8849	0.1653
formatting equal sign	18.52	7.5986	0.0392
formatting vincula	18.41	4.8879	0.1651
formatting equal sign	15.55	4.8963	0.1644
result	2.83	5.1405	0.1457
changing size, position	24.39	4.4906	0.1998
adjusting with Spaces	40.90	3.7449	0.2793
**Total**	**548.36**		**4.8485**

**Table 17 entropy-25-00302-t017:** The steps of solving the problem in the correct *equation* document.

Algorithm	Time (tk)	Ik	Ek
copying equation	7.90	4.7390	0.1775
typing equal sign	3.11	6.0839	0.0897
typing numerator	5.99	5.1383	0.1459
typing denominator	2.91	6.1798	0.0852
typing numerator	4.17	5.6608	0.1119
typing denominator	2.89	6.1898	0.0848
typing numerator	2.95	6.1601	0.0861
typing denominator	2.05	6.6852	0.0650
typing numerator	5.00	5.3989	0.1280
typing denominator	2.12	6.6368	0.0667
typing numerator	4.93	5.4192	0.1266
typing denominator	2.05	6.6852	0.0650
typing numerator	3.00	6.1359	0.0873
typing denominator	3.95	5.7390	0.1075
copying equation	14.89	3.8246	0.2699
changing order	19.00	3.4729	0.3128
copying equation	8.06	4.7100	0.1800
calculating	32.03	2.7195	0.4129
copying equation	9.08	4.5381	0.1953
numerator1 (substitution)	12.85	4.0371	0.2459
numerator2 (substitution)	15.98	3.7226	0.2820
copying equation	16.18	3.7047	0.2841
simplifying fraction1	9.83	4.4236	0.2061
simplifying fraction2	14.06	3.9073	0.2604
result	5.98	5.1407	0.1457
**Total**	**210.96**		**4.2223**

**Table 18 entropy-25-00302-t018:** The steps of adding a new fraction to the formula in the erroneous *equation* document.

Algorithm	Time (tk)	Ik	Ek
typing numerator	11.99	2.2453	0.4736
adjusting with Spaces	11.94	2.2514	0.4728
typing denominator	1.97	4.8509	0.1681
adjusting with Spaces	2.96	4.2635	0.2220
formatting vincula	9.12	2.6401	0.4235
trial and error	9.88	2.5246	0.4387
formatting operator	8.99	2.6608	0.4208
**Total**	**56.85**		**2.6195**

**Table 19 entropy-25-00302-t019:** The steps of adding a new fraction to the formula in the correct *equation* document.

Algorithm	Time (tk)	Ik	
inserting fraction	10.98	0.6374	0.4098
typing numerator	2.91	2.5532	0.4350
typing denominator	3.19	2.4207	0.4521
**Total**	**17.08**		**1.2969**

## Data Availability

Not applicable.

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
