# Peer review of "The Entropy of Digital Texts—The Mathematical Background of Correctness"

_entropy, 2023, doi:10.3390/e25020302_

Round 1

Reviewer 1 Report

Authors propose a study aimed at improving the correcting, formatting, and modification algorithms of documents, measuring the time spent on their modification, and finally calculating the entropy of the completed tasks. The topic of the study is interesting, but the work requires a thorough revision, additional experiments, an expansion of the dataset, and a detailed explanation of the methods chosen. I have some serious observations:

1. Firstly, the abstract is uninformative. You should give specific results achieved and more precise information about the data and methods, so that the reader can understand your contribution to the study.

2. The dataset consists of only 3 documents. Such a sample cannot be considered representative. It is necessary to increase the amount of data for the study or give a strong justification for such a choice.

3. The review of related studies should be put in a separate section and conducted more thoroughly. Use more recent works and modern techniques. Describe the data, methods, and results of the review.

4. Add information about open source datasets—whether they exist and what limitations they have.

5. It should be explained whether the conversion of documents was done manually or automatically.

6. Add a mathematical statement of the problem as a separate section. The metrics that are used to determine the quality of the result should be explained in the form of formulas.

7. A justification should be given as to why these particular methods were chosen.

8. In conclusion, it is necessary to evaluate the statistical significance of the performed transformations.

Author Response

Reply is attached.

Reviewer 2 Report

The manuscript "The entropy of digital texts" presents an application of algorithms previously studied by the authors to calculate the time spent in correcting and formatting three incorrect documents in MS Word formats (DOC and DOCX), in addition to calculating the entropy of information from each completed task. The work is quite exciting and will be of great interest to numerous readers. Overall, the authors present the work accurately, except for the section referring to entropy concepts. Although the manuscript deals well with a subject worthy of investigation, some concepts must be clarified. Please see some comments below.

The title is very general and suggests a review work on the topic under treatment. The present work is a case study. Authors should choose a title that highlights the results of this work.

(a) Given the abundance of entropies proposed in the literature, please clarify in the manuscript that the entropy used was Shannon entropy (information entropy).

(b) The entropy concepts should be removed from subsection "3.5 Recording with ANLITA and calculating entropy". At the same time, a new subsection should be created to explain basic entropy concepts and their connection with the present work. These explanations help the non-specialist in information theory reader better follow the results' analysis. For example, in the results section, the authors analyze by employing the term bits. In this regard, the authors should explain that the choice of base 2 in the logarithm presented in Eq. (4) gives the bit unit for the Shannon entropy.

(c) How does the time spent (on an action) relate to Shannon entropy? Is the longer time related to the greater entropy? This feature is only valid for t_k <= 1/exp(1). The authors should clarify this point in the new version of the manuscript.

(d) The title of the manuscript is quite general and suggests a review work on the topic under study. Indeed, the present work is a case study. Authors should choose a new title that highlights the results of this work.

(e) Please double-check the References. Standardize them. Some references display the journal's name in abbreviated form, while others do not. The first reference must be included in full. Please see the author's guide (https://www.mdpi.com/journal/entropy/instructions).

Author Response

Reply is attached.

Round 2

Reviewer 1 Report

The authors responded to my comments, but not all of them were corrected in the text of the article. The expansion of the review involved not just adding new references but also analytical work with conclusions about the state of the subject area so that the reader could evaluate the contribution of the authors. In addition, you answered my remark regarding the dataset, but you did not add it to the text of the article. correct so that the reader has a better understanding of your work.

Author Response

The manuscript is updated according to the reviewer's suggestion.

Reviewer 2 Report

The authors adequately addressed all questions.
